# Influence of Er- and Al-Doped ZnO on Mixed-Matrix Membranes of Chitosan Derivatives in Bioelectrochemical Systems

**DOI:** 10.3390/molecules30183759

**Published:** 2025-09-16

**Authors:** Santiago Alvear Méndez, Raúl Bahamonde Soria, Daniel Arboleda, Carlos Cevallos, Christian Alcívar, Yessenia Jimenez, Rommy Teran, Henry Pupiales, Patricia Luis

**Affiliations:** 1Renewable Energy Laboratory, Faculty of Chemical Sciences, Central University of Ecuador, Quito 170521, Ecuador; dgarboleda@uce.edu.ec (D.A.); yajimenez@uce.edu.ec (Y.J.); 2Faculty of Chemical Sciences, Central University of Ecuador, Quito 170521, Ecuador; cacevallosm@uce.edu.ec (C.C.); cdalcivar@uce.edu.ec (C.A.); riteran@uce.edu.ec (R.T.); 3Institute of Mechanics, Materials and Civil Engineering-Materials & Process Engineering, University Catholique de Louvain, 1348 Louvain-la-Neuve, Belgium; patricia.luis@uclouvain.be

**Keywords:** bioelectrochemical systems, proton exchange membranes, chitosan derivatives, photocatalyst

## Abstract

Bioelectrochemical systems (BESs) are technologies capable of converting chemical energy into electrical energy or producing value-added compounds. These systems typically employ Nafion membranes as proton exchange separators; however, Nafion is costly and prone to fouling. In this study, mixed-matrix membranes (MMMs) based on chitosan and its derivatives, incorporated with Er- and Al-doped ZnO nanoparticles, were synthesized and evaluated. Key properties assessed included antimicrobial activity, antifouling behavior, chemical stability, water retention, and proton conductivity. The results demonstrated that the chitosan-based membranes doped with Er/Al/ZnO outperformed Nafion in terms of antifouling properties, water retention, and a protective effect on the surface of the membrane against *Escherichia coli* and *Staphylococcus aureus*, while exhibiting comparable proton conductivity and chemical stability.

## 1. Introduction

Proton exchange membranes (PEMs) are essential components in bioelectrochemical systems (BESs), as they enable selective proton transport and maintain separation between anodic and cathodic compartments. Nafion, the most widely used PEM, offers excellent proton conductivity and chemical stability; however, its high cost, limited biodegradability, susceptibility to biofouling, and permeability to gases and organic compounds restrict its long-term use in sustainable or large-scale applications [1].

Given these limitations, the development of alternative membranes that are economical, environmentally friendly, and multifunctional has become a growing priority for the research and development of BES system separators [2]. Mixed-matrix membranes have been attracting the attention of researchers in the field of water treatment, pervaporation, and other applications due to advances in membrane materials, which can improve the selectivity of compounds by exploiting differences in molecular size, polarity, or specific interactions between the membrane and the species of interest. In addition, membranes can have environmental benefits, such as reduced energy consumption and the absence of solvents, which is in line with sustainable industrial practices [3]. However, some of the materials that make up the membranes can be expensive. The use of materials derived from industrial waste, such as chitosan, which can be obtained from the chitin of crustaceans, may be an alternative for the development of membranes [4].

Biopolymers such as chitosan are emerging as attractive candidates for use as separators in bioelectrochemical systems due to their natural origin, biodegradability, antimicrobial properties, and intrinsic functional properties. Chitosan is a deacetylated derivative of chitin, which can be obtained from food industry waste, such as shrimp industry waste [5].

The shrimp industry has an annual production of 5.8 million tons worldwide, with Ecuador being one of its largest producers, with 1.2 million tons in 2022 [6]. Therefore, the use of waste from this industry could be an environmental and industrial contribution in a country such as Ecuador.

Chitosan membranes can have several applications and possibly be used as proton exchange membranes in bioelectrochemical systems [7]. Chitosan membranes reinforced with imidazolate structures can result in high water retention, lower surface resistance, and ionic conductivity comparable to that of the Nafion-117 membrane.

Additionally, chitosan can be chemically modified by replacing the amino group with an imine group, yielding derivatives with improved physicochemical and biological properties. The use of these derivatives as a basis for proton exchange membranes could provide increased antimicrobial and antifouling properties to the separator [5].

Zinc oxide (ZnO) has been reported to confer antifouling and antimicrobial properties to membranes, attributable to the generation of free radicals [8]. This is the case in the study by Vevers et al., which demonstrated that ZnO coupled in ultrafiltration membranes improves wettability, reduces biofouling, and provides antimicrobial activity due to its photocatalytic capacity and the inactivation of more than 99.9% of *Escherichia coli* [9].

Other research incorporating ZnO and SnOx nanoparticles into chitosan biopolymer coatings demonstrated that ZnO photocatalysis improved the thermal stability and hydrophobicity of the material. In addition, a significant reduction in biofouling was also observed [10].

A separate research study on mixed-matrix nanofiber membranes for arsenic filtration indicates that the incorporation of ZnO improves antifouling properties and inhibits the growth of *E. coli* and *S. aureus* [11]. It is worth noting that these modifications improve the chemical and mechanical stability of chitosan membranes, as well as their proton-conducting behavior under acidic conditions. In addition, due to the photocatalytic activity of ZnO, they exhibit antimicrobial and antifouling properties.

However, in order to prevent electron–hole pair recombination and improve photocatalytic efficiency, zinc oxide can be doped with transition metals such as aluminum and erbium.

Aluminum is a good choice for doping due to its small ionic radius, low cost, and availability and because it provides ZnO with better transmittance in the visible region [12]. Additionally, having a 3+ charge and embedding itself into the ZnO structure, it replaces Zn^2+^ ions, generating a charge imbalance that makes the material more electron-donating, thus favoring the generation of electron–hole pairs [13].

Erbium, on the other hand, is stable due to the protective effect of its outer electron shells (5p^6^), and the electronic transitions within its 4f orbitals can cover the entire solar spectrum [14]. This electronic configuration allows Er to trap electrons within the band gap by introducing new energy levels, preventing electrons in the conduction band from recombining with holes in the valence band [15].

Therefore, this study develops and evaluates MMMs composed of chitosan and its Schiff base derivatives, integrated with Er- and Al-doped ZnO nanoparticles. The synthesized membranes are characterized in terms of proton conductivity, water retention, and chemical and mechanical stability, as well as antimicrobial and antifouling performance, to assess their potential as sustainable alternatives to Nafion in BES applications.

## 2. Results and Discussion

### 2.1. Characterization of the Catalyst

#### 2.1.1. Composition of Catalyst

The most efficient doping concentrations of Er and Al were 1.5% and 5%, respectively, with an error margin of ±0.5%, and were obtained from the study conducted by [16]. In that study, undoped ZnO, ZnO doped individually with Er or Al, and ZnO co-doped with both elements were evaluated. It is important to note that, in the present work, we focus exclusively on the synthesis and evaluation of ZnO co-doped with Er and Al. The elemental composition of the synthesized photocatalyst was confirmed by energy-dispersive X-ray spectroscopy (EDX), as shown in Figure 1, where the characteristic peaks of Zn, O, Al, and Er are clearly identified.

#### 2.1.2. Evaluation of Catalyst

It can be observed in Figure 2 that the concentration of methyl orange decreases with time, as at 10 min almost 80% of the dye is removed and at 30 min more than 90%. This could be because the Er/Al/ZnO absorbs the UV radiation, the electrons in the valence band are excited to the conduction band, and electron–hole pairs are formed that participate in the chemical reactions that decompose the dye, which shows that the catalyst works [13].

This phenomenon occurs because Al^3+^ ions can act as electron traps, forming unstable Al^2+^ and Al^4+^ ions that react with oxygen to produce reactive oxygen species responsible for pollutant degradation. Meanwhile, Er^3+^ facilitates the conversion of visible light into ultraviolet light through its 4f electronic levels, enhancing the activation of ZnO under visible light [17].

### 2.2. Evaluation of Membranes

#### 2.2.1. Tensile Test of Membranes

The synthesized membranes were subjected to maximum tensile strength testing following the ASTM D882 standard, applicable to thin plastic films. The measurements were performed using a Shimadzu AGS-X universal testing machine (Shimadzu Corporation, Kyoto, Japan), obtaining an average value of 99.87 N ± 3.77 N, which indicates that the membranes exhibit good mechanical strength suitable for use in bioelectrochemical systems.

#### 2.2.2. Membrane Morphology

The obtained membranes were characterized by scanning electron microscopy (SEM), and the corresponding images are presented in Figure 3. It is observed that the membrane surfaces exhibit a predominantly smooth morphology, without the presence of cracks, excessive porosity, or collapsed structures. This uniform texture indicates that the employed synthesis process was effective in obtaining a continuous and structurally stable polymer matrix.

Additionally, some isolated surface irregularities are identified, likely associated with the incorporation of the photocatalyst. However, these do not compromise the integrity of the membrane. The photocatalytic material particles appear to be well distributed within the chitosan matrix, with no evidence of agglomeration or phase separation (Figure 4), which is indicative of good dispersion and compatibility between the polymer and the photocatalyst. This morphological homogeneity suggests an adequate interfacial interaction between the components, which is fundamental to maintaining the mechanical and functional stability of the membrane during use. Furthermore, the uniform distribution of the photocatalyst may enhance efficiency in photoactivated processes by ensuring homogeneous exposure to the incident radiation field.

#### 2.2.3. FTIR Spectroscopic Analysis of the Membranes

The presence of functional groups in the ZnO/Er/Al powder and in the composite membranes was analyzed by FTIR spectroscopy (Figure 5). In the ZnO samples doped with erbium and aluminum, characteristic bands were observed at 550, 1400, 2000, 2300, and 3300 cm^−1^. The band at 550 cm^−1^ is attributed to the Zn–O stretching vibration, typically located around 430–450 cm^−1^ in pure ZnO, but here it appears shifted to a higher wavenumber due to the incorporation of Er^3+^ and Al^3+^ into the crystal lattice. This shift suggests direct chemical interaction between ZnO and the dopants, indicating that the Er and Al ions were successfully integrated into the zinc oxide structure and modified its local bonding environment. The bands at 1400, 2000, and 2300 cm^−1^ are attributed to carbonates or adsorbed species such as CO_2_, retained due to the high surface reactivity of the oxide.

The characteristic peaks of ZnO/Er/Al also appeared in the FTIR spectrum of the membranes, indicating that the catalyst particles were successfully incorporated into the chitosan polymer matrix. The absence of new peaks suggests that no covalent bonds were formed between ZnO/Er/Al and chitosan, preserving the original chemical structure of both components. However, the shift and broadening of the Amide II band, from ~1550 cm^−1^ to ~1580 cm^−1^, indicates physical interactions such as hydrogen bonding between the amino groups of chitosan and the surface of the nanoparticles, suggesting good compatibility between the polymer and the catalyst.

#### 2.2.4. Antimicrobial Capacity

Figure 6 shows that the MQ, MQ1, and MQ2 membranes provide a barrier against the growth of microorganisms. In the case of *E. coli*, no growth of microorganisms is observed on the surface of the membranes, while for *S. aureus* an inhibition halo of 6 mm is present. Considering that the membrane has a radius of 5 mm, an additional 1 mm is not sufficient to indicate antimicrobial activity. Therefore, it can be said that the membranes do not have a significant inhibitory effect.

However, since there is no bacterial growth on the surface of the membranes, it can be inferred that there is a protective effect against bacterial growth. This could be because the biopolymers form a film on the cell surface that cuts off the passage of nutrients into the cell or also because they are positively charged and can attract the negative charges of bacteria, causing bacterial lysis [5].

Furthermore, it is observed that the MQ1 membrane shows a higher barrier against *S. aureus* compared to the MQ and MQ2 membranes. This could be explained by the fact that when a positive charge is generated on the imine carbon, it can be stabilized by resonance on the aromatic ring.

In contrast, the membranes coupled with the photocatalyst maintain their protective capacity against bacterial growth, as observed in MQF and MQF2. In the case of MQF1, an inhibition halo of 7 mm is presented against *S. aureus*; however, as explained above when analyzing the difference in diameters, this value cannot be considered indicative of antimicrobial activity, although it does suggest a more effective protective barrier. This could be due to the photocatalytic action of the material in the membranes. However, according to [16], Er/Al/ZnO also shows good antibacterial ability even in the dark. Therefore, the antibacterial activity of ZnO nanoparticles against Gram-positive and Gram-negative bacteria could be related to their ability to produce reactive oxygen species (ROS) and to the formation of bonds between metal ions and nitrogen, oxygen, or sulfur atoms in biological molecules [17].

#### 2.2.5. Water-Holding Capacity

Among the membranes without a photocatalyst, the one with the highest water-holding capacity is MQ, with a value of 60.22%, probably because chitosan has hydroxyl and amine groups that can form hydrogen bridges with water and therefore absorb more water [18]. However, with the chitosan-derived membranes, MQ1 and MQ2, with 58.02% and 54.77%, respectively, the percentage decreases because the amount of hydrogen bridges probably decreases. In the case of the MQ1 membrane, this is likely because the amine group is transformed to an imine group, and in the case of the MQ2 membrane, this is likely because of the presence of the methoxy group.

In contrast, the effect of Er/Al/ZnO is more pronounced in the membrane with an MQF photocatalyst, while in MQF1 and MQF2 its impact is lower, with values of 46.95%, 52.79%, and 51.09%, respectively. This decrease in the water retention capacity may be due to the fact that Er/Al/ZnO interacts with the -OH and -NH_2_ groups of the chitosan of the MQF membrane, forming a chitosan–Er/Al/ZnO complex and preventing the formation of hydrogen bonds with water. This would not occur with MQF1 and MQF2 membranes, as the coordination between Zn and N could be complicated by the formation of the imine functional group, resulting in a minor impact compared to MQF [19].

Despite the reduction in this property, membranes with and without a photocatalyst have a higher water retention percentage than a Nafion separator, which has a value of 21.8% water retention. This is beneficial because higher water retention improves other membrane properties such as antifouling and proton conductivity.

#### 2.2.6. Chemical Stability

Figure 7a shows that the membranes in general have a high percentage of hydroxyl tolerance, which shows that they have a high tolerance to strong oxidizing conditions, as they resisted the Fenton reagent for a period of 7 days [20].

The values of 91.32%, 91.10%, and 90.28% for MQ, MQ1, and MQ2 indicate that each membrane retains its minimum weight up to 90%; however, the values of 94.73%, 93.06%, and 95.61% for MQF, MQ1F, and MQ2F, respectively, suggest that Er/Al/ZnO improves the chemical stability of the membranes, probably because the photocatalyst coordinates with the biopolymer by blocking the reactive groups of MQF, MQF1, and MQF2 [17]. Furthermore, compared to Nafion membranes with 100% stability, the tested membranes have a lower tolerance to hydroxyls; however, the membrane performance is close to that of Nafion membranes.

#### 2.2.7. Fouling

To determine this property, the surface electrical resistance of the membranes with and without a photocatalyst was measured (Figure 7b). The values obtained in the fouling test were 11.03 Ω·cm^2^, 10.08 Ω·cm^2^, and 8.07 Ω·cm^2^ for MQ, MQ1, and MQ2, respectively. These values were higher than those obtained for MQF, MQ1F, and MQ2F, which were 6.84 Ω·cm^2^, 5.15 Ω·cm^2^, and 5.06 Ω·cm^2^.

This suggests that when membranes are coupled with Er/Al/ZnO, ionic conductivity is enhanced [21]. Although the photocatalyst may decrease water retention within the membrane, its -OH groups, formed on the Er/Al/ZnO surface, interact with water molecules. Thus, a water film would form on the membrane surface, preventing organic matter and mineral salts from accumulating on the membrane [22].

In addition, the improvement in antifouling behavior is also considered to be due to the photocatalytic properties conferred by erbium and aluminum to ZnO, enabling the degradation of organic contaminants on the membrane surface under light conditions [16].

Likewise, the literature mentions that the addition of Er/Al/ZnO reduces the pore size of a membrane, so intuitively one would think that it would increase the resistance to the passage of ions. However, the results show the opposite. This indicates that the passage of ions is not due to the membrane pores but would be attributed to the -OH groups of ZnO, i.e., the hydrophilicity of the membrane [23].

Similarly, Nafion membranes in the fouling test have a surface electrical resistance of 5.31 Ω·cm^2^, with the MQ1F and MQ2F photocatalyst membranes being the ones that can be compared to Nafion membranes.

#### 2.2.8. Biofouling

In this test, the behavior was similar to the previous one, since the values 13.09 Ω·cm^2^, 11.39 Ω·cm^2^, and 8.89 Ω·cm^2^ for MQ, MQ1, and MQ2, respectively, are higher than 8.06 Ω·cm^2^, 6.03 Ω·cm^2^, and 5.94 Ω·cm^2^ for MQF, MQ1F, and MQ2F. This decrease in surface electrical resistance suggests that the photocatalyst, Er/Al/ZnO, enhances the ionic conductivity in the membranes [24].

This may be due to the antibacterial properties and photocatalysis provided by Er/Al/ZnO to the membranes. This is first because the waste products and metabolites of bacteria would not accumulate on the membrane surface and second because all the accumulated organic and inorganic matter would be oxidized due to the hydroxyl radicals generated in the process of photocatalysis [23].

Finally, in this test, the Nafion membranes fouled 10.21 times more, while the tested membranes fouled less than two times. This indicates that the chitosan-derived and Er/Al/ZnO membranes have better antifouling properties than the Nafion membranes, so the analyzed membranes could maintain their efficiency in environments where biofouling may exist.

#### 2.2.9. Proton Exchange Capacity

In Figure 8a,b, the impedance spectra, which were simulated with the equivalent circuit 8c, suggest that at high frequencies the electrolyte resistance at the membrane–electrolyte interface and at low frequencies the charge transfer and diffusion of species takes precedence [25].

Although the measured conductivity of Nafion was lower than the typical value reported in the literature (~0.1 S/cm), all membranes were tested under the same experimental conditions. Therefore, the results are still valid for comparing their relative performance. The charge transfer value of the Nafion membrane indicates that it may have better proton exchange capacity than chitosan membranes and their derivatives. However, the charge transfer values in Figure 9 of the MQF, MQ1F, and MQ2F membranes suggest that the photocatalyst gives the spacers improvements in proton affinity as they increase their conductivity by 1.23, 2.27, and 1.67 times, respectively, compared to the membranes without Er/Al/ZnO. In the MQ1F membrane, with a conductivity value of 21.250 μS/cm, the greatest effect of the catalyst is observed, and the MQ2F membrane, with a value of 21.250 μS/cm, has a conductivity more similar to that of Nafion, surpassed by the latter by only 1.83 times. This may be due to the -OH groups found on the zinc oxide outer groups and through the Grotthus mechanism transferring protons due to the formation of hydrogen bridges between the water and the catalyst surface [26].

## 3. Material and Methods

### 3.1. Materials and Reactives

Chitosan (Q) and derivatives, Q1 (chitosan + salicylaldehyde) and Q2 (chitosan + methoxybenzaldehyde), were synthesized. Sulfuric acid (H_2_SO_4_, 98%), hydrochloric acid (HCl, 37%), sodium chloride (NaCl, 99.99%), anhydrous sodium sulfate (Na_2_SO_4_, ≥ 99.0%), bovine serum albumin (BSA lyophilized powder ~66 kDa), ferrous sulfate heptahydrate (FeSO_4_-7H_2_O, 99.5%), hydrogen peroxide (H_2_O_2_, 30%), and acetic acid (CH_3_COOH, 99.8%) were acquired from Merck KGaA (Darmstadt, Germany). Additionally, zinc nitrate hexahydrate (Zn(NO_3_)_2_-6H_2_O, 99%), oxalic acid dihydrate ((COOH)_2_-2H_2_O), absolute ethanol (CH_3_CH_2_OH, ≥99.5%), aluminum nitrate nonahydrate (Al(NO_3_) _3_-9H_2_O, ≥98%), and erbium nitrate pentahydrate (Er(NO_3_)_3_-5H_2_O, 99.9%) were supplied by Sigma-Aldrich (St. Louis, MI, USA). Mueller–Hinton agar and the antibiotic discs of ciprofloxacin, along with the bacterial strains *Escherichia coli* ATCC 25922 and *Staphylococcus aureus* ATCC 25923, were obtained from the Microbiology Laboratory of Chemical Sciences at the Central University of Ecuador (Quito, Ecuador). Finally, the PZ-FSR-1-DRY ultrafiltration membranes made of polyacrylonitrile (PAN) were obtained from Synder Filtration (Vacaville, CA, USA).

### 3.2. Chitosan Synthesis

Shrimp shells were allowed to dry at room temperature, and then their particle size was reduced. Next, 1 kg of shrimp shell powder was added to a 2 M HCl solution using a 1:5 (*w*/*v*) ratio and stirred for 2 h at room temperature. Then, it was washed with distilled water until a neutral pH was obtained, and the solid obtained was dried in an oven at 65 °C for 24 h and crushed. Subsequently, the product obtained was added to a 2 M NaOH solution, stirred at 100 °C for 3 h, allowed to dry in an oven at 65 °C for 24 h, and then the sample was crushed. Finally, the product was refluxed with a 50% NaOH solution in a 1:10 (*w*/*v*) ratio at 120 °C for 22 h and washed to a neutral pH; then, the product was dried in an oven at a temperature of 65 °C for 24 h, and the sample was crushed [27].

### 3.3. Synthesis of Chitosan Derivatives

For the synthesis of chitosan Schiff bases, a given amount of chitosan is dissolved in a 2% acetic acid solution, and ethanol is added and reacted with an aldehyde or ketone. In the case of the synthesis of the first derivative, chitosan is reacted with salicylaldehyde under reflux for 8 h, while, for the second derivative, chitosan is reacted with methoxybenzaldehyde and stirred for 8 h at room temperature. Finally, the product obtained is washed with ethanol and allowed to dry [28].

### 3.4. Synthesis of Er/Al/ZnO

Two solutions were prepared. In solution A, 0.005 mol of zinc nitrate hexahydrate was dissolved in 30 mL of EtOH and stirred at 60 °C for 30 min; then, 0.175 g of aluminum nitrate and 0.066 g of erbium nitrate nonahydrate were added and stirred for an additional 30 min. In contrast, solution B was prepared by dissolving 0.03 mol of oxalic acid dihydrate in 40 mL EtOH and stirred at 50 °C for 30 min. Then, solution B was added over solution A dropwise for 1 h, dried at 80°C in the oven for 1 day, and calcined in the muffle for 4 h at a temperature of 400 °C [29]. A scanning electron microscope, Carl Zeiss equipment (Carl Zeiss Microscopy GmbH, Oberkochen, Germany), and a BALZERS SCD 030 coater (Balzers Union, Balzers, Liechtenstein) were used to characterize the catalyst.

### 3.5. Evaluation of Photocatalytic Capacity

A solution of methyl orange with a concentration of 10 ppm was prepared, to which Er/Al/ZnO was added at a concentration of 0.03% w/w. To this solution, under constant agitation and in total darkness, white light was irradiated by means of a 20 W lamp, and then the absorbance of the solution was measured every 10 min with a wavelength of 469 nm, with first centrifuging for 5 min. The following formula was used to calculate the percentage of methyl orange removal:$$\%Remoción=\frac{A_{0}-A_{t}}{A_{0}}\times 100\%$$
where *A*_0_ is the initial absorbance and *A_t_* is the absorbance over time [21].

### 3.6. Preparation of Mixed-Matrix Membranes

The mixed-matrix membranes were prepared by the pour-through method on an ultrafiltration membrane (PAN) following the methodology of [4]. The membranes prepared without a photocatalyst have the following abbreviations: chitosan membranes (MQ), Q1 membranes (MQ1), and Q2 membranes (MQ2). In contrast, the membranes with a photocatalyst have the following abbreviations: chitosan membranes (MQF), Q1 membranes (MQ1F), and Q2 membranes (MQ2F).

### 3.7. Membrane Morphology

The morphology of the membranes was analyzed using scanning electron microscopy (SEM) with a Carl Zeiss instrument (Carl Zeiss Microscopy GmbH, Oberkochen, Germany). To obtain clean cross-sections, the samples were previously fractured in liquid nitrogen. Subsequently, they were coated with a thin layer of gold by vacuum sputtering using a BALZERS SCD 030 coater (Balzers Union, Balzers, Liechtenstein) in order to provide electrical conductivity. Meanwhile, the FT-IR spectra of ZnO/Er/Al and the membranes were obtained using a Bruker spectrometer from 400 cm^−1^ to 4000 cm^−1^ in ATR mode to investigate the chemical properties of all membrane samples.

### 3.8. Antimicrobial Capacity

Membrane samples with a radius of 5 mm were tested both without a photocatalyst (MQ, MQ1, and MQ2) and with photocatalyst (MQF, MQ1F, and MQ2F). Prior to testing, all materials, including Mueller–Hinton agar and 0.9% saline, were sterilized. Bacterial suspensions of Staphylococcus aureus and Escherichia coli were prepared in sterile saline to a turbidity equivalent to a 0.5 McFarland standard and inoculated evenly onto the agar surface in Petri dishes. Ciprofloxacin was used as a positive control, while 2% acetic acid served as a negative control. The plates were incubated at 35 °C for 24 h, after which the presence or absence of bacterial growth on the membrane surface was visually assessed [8].

### 3.9. Water-Holding Capacity

This property was measured by gravimetry, measuring the mass of the wet and dry membranes. The following formula was used to calculate the water-holding capacity of the membranes:
$$X=\frac{W_{f}-W_{0}}{W_{0}}\times 100\%$$
where *X* is the percentage of water retention, *W_f_* is the weight of the wetted membrane, and *W*_0_ is the weight of the dry membrane [30,31].

### 3.10. Chemical Stability

Chemical stability is the ability of a membrane to resist severe oxidative conditions, and this property was measured based on the methodology of [20]. The following formula was used to determine the tolerance of hydroxyl radicals.
$$T=\frac{W_{d}-W_{a}}{W_{d}}\times 100\%$$
where *T* is the tolerance of hydroxyl radicals and *W_a_* and *W_d_* are the masses in g of the membranes when dry and after being attacked by hydroxyl radicals.

### 3.11. Fouling and Biofouling

For this procedure, the surface electrical resistance was measured. For the fouling tests, a four-compartment cell filled with 0.2 M sodium sulfate and 0.1 M sodium chloride was used. In contrast, for the biofouling tests, the same cell was used but with the addition of bovine serum [32]. The following formula was used to calculate the surface resistance:
$$R_{s}=\left(\frac{V-V_{0}}{I}\right)\times A$$
where *V* is the voltage of the membrane to be studied, *V*_0_ is the target voltage, *I* is the current (0.011 A), and *A* is the effective area (7.065 cm^2^) of the membrane.

### 3.12. Proton Exchange Capacity

For this procedure, the electrochemical impedance spectroscopy technique was used from a frequency of 10 Hz to 50 KHz. For this purpose, a 0.01 M HCl solution, two Ag/AgCl electrodes, and a potentiostat were used [33]. The following equation was used to calculate the proton conductivity:
$$\sigma =\frac{d}{Rct\times A}$$
where σ is the proton or salt conductivity and *d* is the membrane thickness (0.016 cm). *R_ct_* is the charge transfer resistance and *A* is the surface area of the membrane (2.759 cm^2^).

## 4. Conclusions

In this research work, mixed-matrix membranes of chitosan and chitosan derivatives coupled with a ZnO photocatalyst doped with Er and Al were synthesized. These membranes showed higher water retention than Nafion, high chemical stability under oxidative conditions, and improved antifouling and antibiofouling behavior compared to membranes without a photocatalyst. Additionally, a protective effect against the growth of *Escherichia coli* and *Staphylococcus aureus* was evidenced, which was maintained or increased with the addition of the photocatalyst, along with a proton exchange capacity comparable to Nafion membranes, which increased up to twice its value when the photocatalyst was added.

The synthesized and studied membranes have great potential for use in bioelectrochemical systems, as in some properties they outperform Nafion membranes and in others they are comparable. Therefore, future studies could be carried out under real operational conditions in bioelectrochemical systems to analyze their durability and performance, as well as to make modifications that enhance their properties.

## Figures and Tables

**Figure 1 molecules-30-03759-f001:**
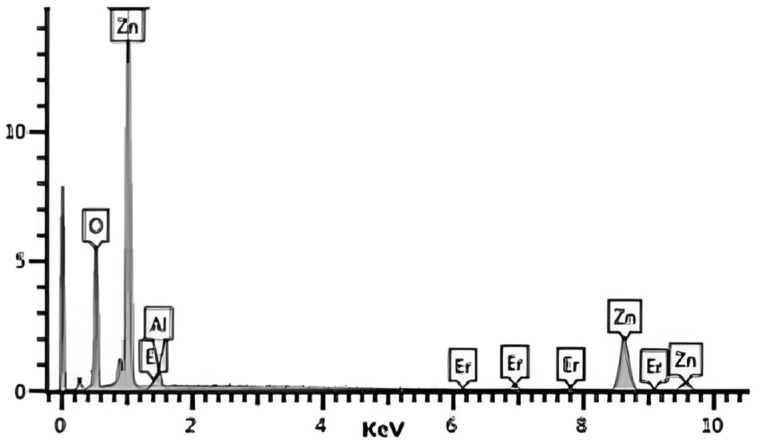
Energy-dispersive X-ray spectroscopy (EDX) spectrum of the ZnO photocatalyst co-doped with erbium (Er) and aluminum (Al). Characteristic peaks of Zn, O, Al, and Er are identified, confirming the successful incorporation of the dopant elements into the zinc oxide structure.

**Figure 2 molecules-30-03759-f002:**
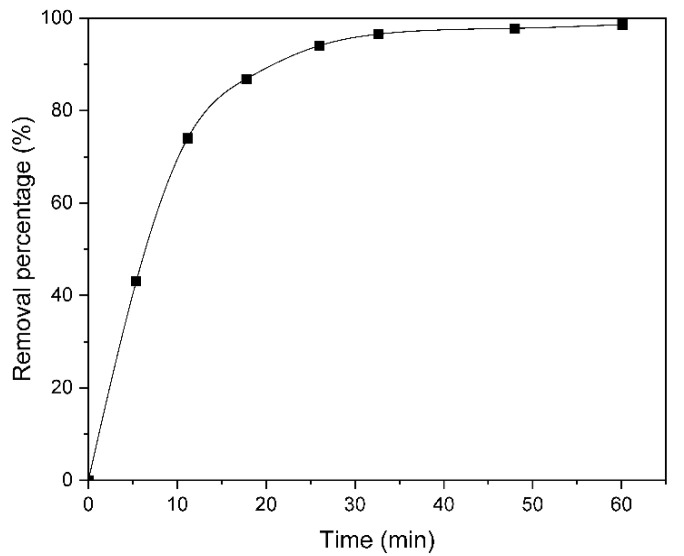
Curve of the removal percentage of 10 ppm methyl orange as a function of time, using 0.03% w/w Er/Al/ZnO as a photocatalyst.

**Figure 3 molecules-30-03759-f003:**
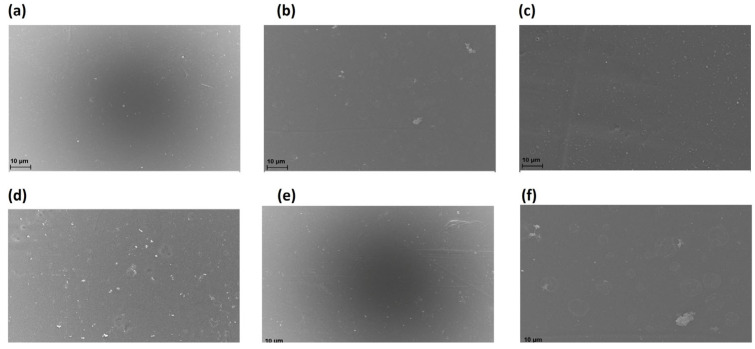
SEM images of the top surface of the membranes: (**a**) MQ, (**b**) MQ1, (**c**) MQ2, (**d**) MQF, (**e**) MQ1F, and (**f**) MQ2F. The observed morphological variations are related to the incorporation of the photocatalyst in the different chitosan derivatives.

**Figure 4 molecules-30-03759-f004:**
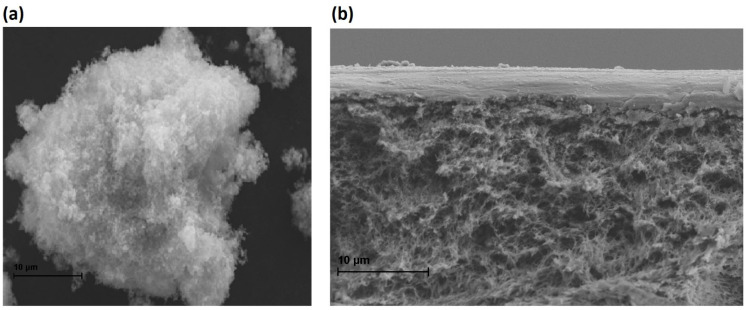
(**a**) SEM image of the ZnO/Er/Al photocatalyst showing its characteristic morphology. (**b**) SEM image of the cross-section of a composite membrane, obtained by overlaying a PAN layer and a chitosan layer functionalized with ZnO/Er/Al.

**Figure 5 molecules-30-03759-f005:**
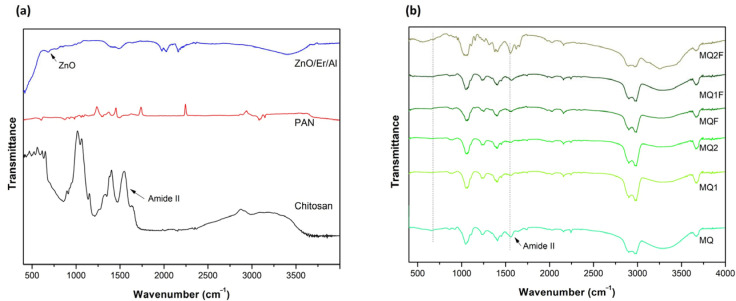
(**a**) FTIR spectra of the PAN polymer, chitosan, and the ZnO/Er/Al catalyst used as base materials. (**b**) FTIR spectra of the membranes prepared with chitosan derivatives and the incorporation of the ZnO/Er/Al photocatalyst.

**Figure 6 molecules-30-03759-f006:**
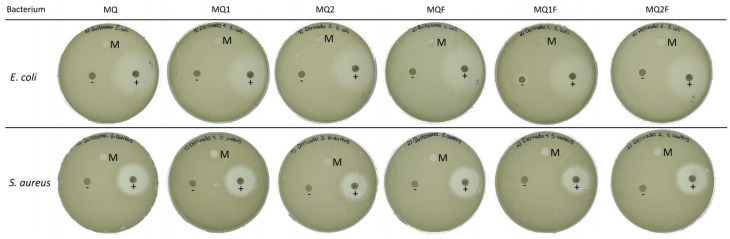
Disk diffusion test on membranes with and without photocatalyst against Staphylococcus aureus and Escherichia coli: (M) membrane sample; (+) positive control; (−) negative control.

**Figure 7 molecules-30-03759-f007:**
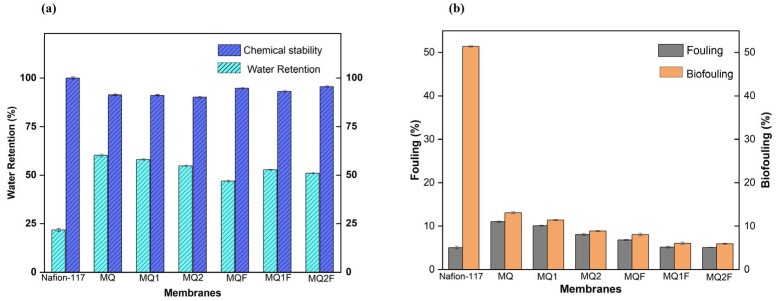
(**a**) Water retention capacity and chemical stability under oxidizing conditions in membranes with and without a photocatalyst; (**b**) evaluation of fouling and biofouling on membranes with and without a photocatalyst.

**Figure 8 molecules-30-03759-f008:**
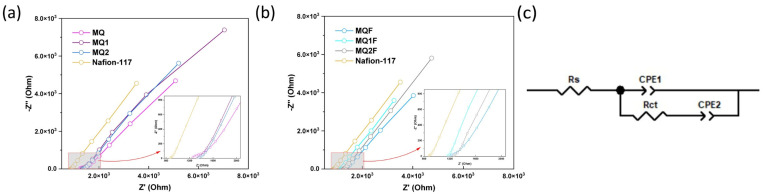
(**a**) Impedance spectra of membranes without photocatalyst; (**b**) impedance spectra of membranes with photocatalyst; (**c**) circuit equivalent.

**Figure 9 molecules-30-03759-f009:**
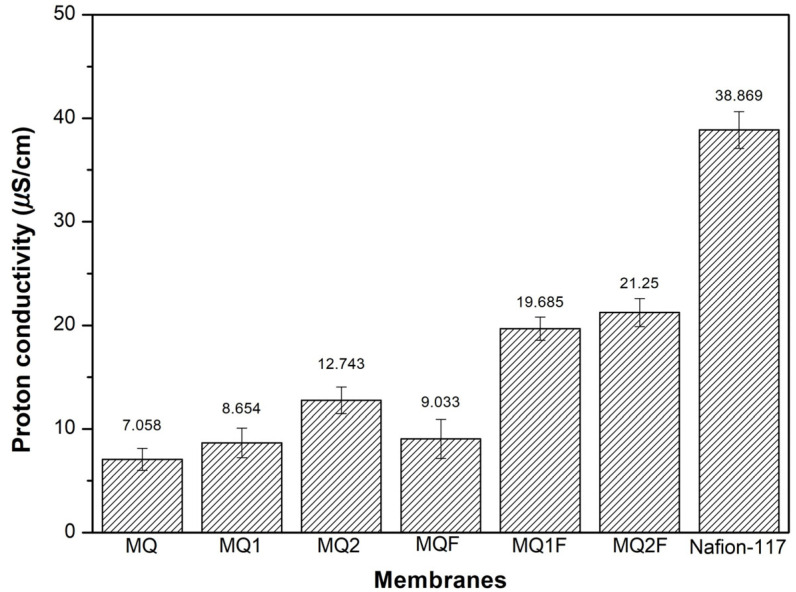
Proton conductivity of membranes without photocatalyst, with photocatalyst, and Nafion.

## Data Availability

Data are contained within the article.

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
