# Peer review of "Influence of Er- and Al-Doped ZnO on Mixed-Matrix Membranes of Chitosan Derivatives in Bioelectrochemical Systems"

_molecules, 2025, doi:10.3390/molecules30183759_

Round 1
Reviewer 1 Report
Comments and Suggestions for Authors
In this study, mixed matrix membranes (MMMs) based on chitosan and its derivatives, incorporated with Er- and Al-doped ZnO nanoparticles, were synthesized and evaluated. The results demonstrated that the chitosan-based membranes doped with Er/Al/ZnO outperformed Nafion in terms of antifouling properties, water retention, and a protective effect on the surface of the membrane against Escherichia coli and Staphylococcus aureus, while exhibiting comparable proton conductivity and chemical stability.Overall, the authors have conducted substantial research, and their work has value in the field of bioelectrochemical systems. However, there are still some issues that need to be addressed:
- The introduction lacks a clear comparison with recent similar studies, and the specific advantages and mechanism of Er- and Al-doped ZnO are not well articulated. Please include a comparison with recent literature (last 3 years) to highlight the novelty and improvement of your material design.
- Some procedural details in the materials synthesis and testing sections are insufficient, which may affect reproducibility. For instance, Section 3.3 does not specify the doping ratios of Er3+ and Al3+ relative to Zn2+. Please clarify these details and briefly state the error margins at the end of each method section to ensure methodological transparency.
- The antimicrobial mechanism of Er/Al/ZnO lacks solid experimental support. No comparison was made between dark and light conditions, and the individual effects of Er-ZnO and Al-ZnO were not tested. Please add appropriate control experiments and strengthen the discussion with supporting literature.
- The figure descriptions in Section 4 are too brief and do not adequately explain the data trends or their relevance to the study’s hypotheses. In particular, the captions for Figures 1, 2, and 3 lack sufficient detail, and the discussion fails to link the visual data to the proposed mechanisms.
- Sections 4.2–4.3 present the results but lack sufficient comparison with existing literature. It is recommended to compare the findings with key references to highlight the synergistic effects of Er/Al doping.
Author Response
Thank you for your comments. I am attaching the file with responses to all the observations made.

Reviewer 2 Report
Comments and Suggestions for Authors
The current study developed mixed matrix membranes (MMMs) based on chitosan and incorporating Er- and Al-doped ZnO nanoparticles, which exhibited antimicrobial activity, antifouling behaviour, chemical stability, water retention, and, notably, proton conductivity. These proposed membranes outperformed Nafion Membranes. The manuscript is well organized, and the results are sound. However, the following comments can be addressed before publication.
- Introduction section: This section lacks a description of other current approaches (i.e., state-of-the-art), including advantages, drawbacks, as well as performances (numbers).
- There is no section 2 in the manuscript (it jumps from section 1 (introduction) to section 3-Materials)
- The reviewer suggests merging Materials and Methods into one section.
- Figure 2: Increase the resolution of the Figure (too blurry).
- Figure 3a, include error bars.
- Figure 4, too small to notice any detail, please increase the size
- Turn Table 1 into a plot.
- The conclusions section is a mere repetition of results. Please include future perspectives and implications.
Author Response

(The authors gave the same response as above.)

Reviewer 3 Report
Comments and Suggestions for Authors
This study developed chitosan derivative-based mixed-matrix membranes (MMMs) applicable to bioelectrochemical systems (BESs) and experimentally analyzed the enhancement of the physicochemical properties, antimicrobial activity, fouling resistance, and conductivity of the membranes by introducing Er and Al-doped ZnO nanoparticles. The core significance of this study lies in its exploration of the potential of using inorganic doping materials on a naturally occurring polymer as a replacement for Nafion, demonstrating both its high practical and environmental value. However, the following points require further improvement for publication in Molecules:
- While the rationale for selecting Er and Al-doped ZnO is briefly mentioned, there is a lack of a clear theoretical basis or citations from prior research regarding the expected property enhancements in BESs (e.g., increased proton conductivity, electron mobility, and enhanced antimicrobial activity). Please provide a more systematic introduction to the effects of Er/Al doping on the band gap structure, conductivity, and antimicrobial activity of ZnO.
- Chapter 2 is not included in the text. Perhaps “3. Materials,” should be revised to “2. Materials.”
- Analytical results, such as FT-IR, should be included to confirm the chemical structure of the prepared MMMs. Additionally, FE-SEM images that can verify the morphological characteristics of the MMMs should be provided.
- The resolution of Figure 2 is too low, making it difficult to identify sample information. Please consider enhancing the image quality.
- Tensile stress-strain curves that can determine the physical strength of the prepared MMMs should be measured and the data presented.
- A schematic diagram illustrating the operating principle of the BES to be applied in this study should be added. Furthermore, the results of performance evaluations of the fabricated membranes applied to a BES should be presented.
- The conclusion should highlight the key findings of this study in more detail to better emphasize its contributions.
Author Response

(The authors gave the same response as above.)

Round 2
Reviewer 1 Report
Comments and Suggestions for Authors
The manuscript would benefit from thorough language polishing throughout to enhance clarity and fluency.
Reviewer 3 Report
Comments and Suggestions for Authors
The authors have carefully revised the manuscript according to the referees’ comments. In my opinion, this manuscript could be accepted for publication in Molecules.